# One-Step Laparoscopic Abomasopexy for Left Displacements of the Abomasum in Pregnant Cattle: A Retrospective Study

**DOI:** 10.3390/ani12233264

**Published:** 2022-11-23

**Authors:** Hideo Iso, Fumikazu Uchiyama, Takeshi Tsuka

**Affiliations:** 1Iso Veterinary Service, 451–14 Shimakata, Nasushiobara 329-3152, Japan; 2Clinical Veterinary Sciences, Joint Department of Veterinary Medicine, Faculty of Agriculture, Tottori University, 4-101, Koyama-Minami, Tottori 680-8553, Japan

**Keywords:** Holstein cattle, left abomasal displacement, one-step laparoscopic abomasopexy, pregnant, toggle–pin–suture

## Abstract

**Simple Summary:**

One-step laparoscopic abomasopexy contributed to satisfactory outcomes in all 15 cows affected by left displacements of the abomasum during late pregnancy. This technique allows for appropriate correction of the left-displaced abomasum without being hindered by the gravid uterus. The treated cows had no postoperative complications prior to parturition and had good reproductive performance. Nine of the 15 animals were successfully bred over one year after the operation.

**Abstract:**

Although technical descriptions have been published, the applicability of left-flank laparoscopy for the surgical correction of left displacement of the abomasum (LDA) in pregnant cattle has not yet been fully assessed. This study aimed to investigate the postoperative performance of one-step laparoscopic abomasopexy for the correction of LDA during late pregnancy. Fifteen pregnant Holstein cattle with LDA were treated with one-step laparoscopic abomasopexy between days 259 and 274 of gestation. This technique was performed in a standing position. Under endoscopic observation via trocars introduced from the left flank, the animals underwent a surgical procedure to place the bar part of a toggle–pin–suture (TPS) device into the lumen of the abomasum through a long cannula. A suture of a TPS device was secured to the ventral abdominal wall by using a long applicator. No cases experienced postoperative complications, and all had a normal delivery on postoperative day 17.4 ± 4.9. Three of the 15 animals exhibited foot diseases and mastitis after calving. The reproductive performance was recorded during lactation periods, showing a service conception rate of 2.9 ± 1.2 and 110.5 ± 39.1 open days. The one-year survival rate was 60% (9/15). The positive postoperative results demonstrate the benefits of one-step laparoscopic abomasopexy in pregnant bovine cases with LDA.

## 1. Introduction

Displacement of the abomasum is one of the most common peripartum diseases in bovines, occurring frequently between two weeks prepartum and postpartum [1]. Of the animals that develop this disease, 90% present within six weeks postpartum, predominantly during the first week [1,2,3]. Displacement of the abomasum can also occur in 2–10% of pregnant cows, mostly within the last three weeks before calving [2]. On the other hand, the periods of onset in heifers were reported to be between four and five months of gestation, despite unknown etiological factors [2,4]. Most previously reported pregnant cases involved left displacement of the abomasum (LDA) [5]. Late pregnancy may be considered one of the predisposing factors in the development of LDAs, given that abomasal atony is associated with a nervous reflex and the rearrangement of the abdominal viscera [5]. The gravid uterus forces the abomasum forward into a semi-displaced position [1,2,5]. Prepartum occurrence of LDAs can be caused by excess accumulation of gas and ingesta in the lumen of the semi-displaced abomasum [1]. Mechanical outflow disturbance is one of the minor causes of LDAs in pregnant cows, together with adhesive peritonitis that is associated with a foreign body and twin gravid uterus [2,6]. Although prepartum LDAs are commonly asymptomatic, pregnant cases with LDAs that present with clinical signs, such as loss of appetite, tympany, and constipation, require surgical intervention [2,4,6,7,8].

The casting and rolling technique has been conducted as a conservative therapeutic option for correcting LDAs [9,10,11]. Despite being used since the 1960s, laparotomy remains one of the most widely performed therapeutic techniques for correcting LDAs [12]. Common laparotomy techniques for correcting LDAs include left-flank abomasopexy, right-paramedian abomasopexy, and right-flank omentopexy (and/or pyloropexy) [9,13]. The blind suture and toggle–pin–suture (TPS) techniques have been used as minimally invasive techniques in the correction of LDAs since the 1970s and 1980s, respectively [13,14]. Advanced, minimally invasive surgical techniques have been employed in laparoscopy since the 2000s, allowing real-time assistance for the blind repositioning of the dislocated abomasum, such as in the roll and tack technique and TPS [12,13]. The two-step technique signifies laparoscopy-assisted abomasopexy by using the TPS technique while the animal is positioned in dorsal recumbency, subsequent to standing [13]. The one-step technique allows for abomasopexy by using the TPS technique under laparoscopic guidance via the left paralumbar area in a standing position [12]. 

The rolling technique [11], left-flank laparotomy [2], and two-flank laparotomy [4,6] have previously been performed on pregnant cows with LDAs. In terms of the previous application of laparoscopic abomasopexy in pregnant cases, the one-step technique has been applied in a single case, though the process was not described in detail [15]. This report aimed to examine the therapeutic efficacy of one-step laparoscopic abomasopexy when applied to pregnant bovine LDA cases and to discuss the reasons for its favorable outcomes.

## 2. Materials and Methods

### 2.1. Animals

The subjects were 15 Holstein milking cattle reared at 11 dairy farms that commonly presented with a sudden decrease in or total loss of appetite during late pregnancy. In all cases that were examined by auscultation with percussion over the left intercostal and flank areas, a ‘ping’ sound was heard between the left tenth and thirteenth intercostal spaces. LDAs were diagnosed based on these clinical symptoms.

### 2.2. Procedure of One-Step Laparoscopic Abomasopexy for Pregnant Cattle

Each animal was kept in a standing position, and xylazine hydrochloride (0.05 mg/kg) was administered for mild sedation prior to the preoperative analgesia of flunixin meglumine (2 mg/kg); both were administered through intravenous injection. The entire left flank was then clipped and disinfected. Local anesthesia consisted of a subcutaneous injection of procaine hydrochloride. Surgery was performed with laparoscopic abomasopexy equipment (FRIGZ Medical Japan Co., Ltd., Chiba, Japan) comprising a 29-cm-long endoscope (49 mm outer diameter), a halogen light source, and a long applicator, needle, and cannula designed to operate between the left flank and the paramedian area of the ventral surface of the abdomen. Two types of trocar cannulas (one was 13 mm in outer diameter and 11 cm in length, and the other was 14 mm in outer diameter and 11 cm in length; Olympus Co., Tokyo, Japan) were used for the first and second ports, respectively.

A first small incision was made in the upper area of the left flank close to the thirteenth rib. A first trocar was subsequently introduced into the abdominal cavity through an incision. The gravid uterus was not commonly observed in the intra-abdominal view of the endoscope passing through the first trocar because it was located behind the rumen in the caudal abdomen. A distended abomasum was commonly detected in the space between the surface of the rumen and the left abdominal wall when the endoscope was turned downward and toward the front (Figure 1A). Under intra-abdominal observation from the endoscope turned upward and toward the front, a second trocar was introduced into the 11th and 12th intercostal spaces (Figure 1B). A 39-cm-long cannula (5 mm in outer diameter) was inserted into the lumen of the upward-dislocated, distended abomasum through centesis, with the sharp apex of the cannula passing through the second trocar (Figure 1C). The bar part of a TPS device (Japan Paramedic Co., Ltd., Tokyo, Japan) was then inserted into the lumen of the abomasum through the cannula. The gas spontaneously escaped from the lumen of the abomasum through the cannula. Under endoscopic observation, it was possible to decompress the abomasum and return it to its original position. The apex of the cannula was subsequently pulled out from the lumen of the abomasum (Figure 1D). 

After the removal of the cannula from the second trocar, a suture part of the TPS device was picked up from the second trocar. The next procedure was prepared by using a suture part of a TPS device tied to the apex of a 130-cm-long needle (7 mm in outer diameter) and set into a 115-cm-long applicator (7 mm in inner diameter and 8 mm in outer diameter) (Figure 2A). The applicator was introduced through the second trocar into the abdominal cavity, while the apex of the needle was withdrawn from the applicator to prevent injury to the abdominal organs (Figure 2B). Under endoscopic observation, the apex of the applicator was advanced along the surface of the rumen toward the ventral abdominal wall. When the apex of the applicator could reach the right side of the midline of the abdomen, the apex of the long needle protruded outside the ventral abdominal wall (Figure 2C). The suture of the TPS device was then picked up and tied over a gauze stent by an assistant veterinarian (Figure 2D). The one-step laparoscopic abomasopexy procedure was successfully performed within 15–40 min. Antibiotics were not administered postoperatively.

## 3. Results

The animals were aged 1968.2 ± 431.6 (783–2604) days and experienced 2.5 ± 1.1 (0–5) parity when treated with one-step laparoscopic abomasopexy between 2015 and 2018 (Table 1). The gestation periods at the time of surgery were 266.9 ± 4.8 (259–274) days.

The clinical signs resolved within one to two postoperative days in all 15 pregnant LDA cases that were treated with one-step laparoscopic abomasopexy. No postoperative complications, such as infection or abortion, were observed. These animals were all able to have a normal, single-birth delivery at 17.4 ± 4.9 (10–27) days postoperatively (Table 2). In three of the 15 subjects, arthritis, mastitis, and sole ulcers were recorded postpartum. During the lactation periods following the prepartum surgical repair of LDA, conception could be achieved in 13 of 15 animals, with the two exceptions being planned to be culled during this lactation period. Of the 13 subjects who conceived again, services were required 2.9 ± 1.2 (1–5) times for conception in 12 animals, with no record of conception service for one animal. The days open were 110.5 ± 39.1 (63–181). During the periods of lactation under observation, six animals were culled between 83 and 157 days after the operation; the causes for culling in three animals were hip dislocation, sole ulcers, and poor milk productivity, with unknown reasons in three animals. Two additional animals were culled premeditatedly on 443 and 420 postoperative days due to periarthritis and poor milk quality, respectively. The remaining seven animals were kept lactating between 369 and 962 postoperative days. The one-year survival rate was 60% (9/15). In all 15 cases, no recurrence of abomasal displacement was observed during this study’s periods.

## 4. Discussion

Based on the clinical records shown in Table 1, fifteen multiparous Holstein cows were treated with one-step laparoscopic abomasopexy, typically during late pregnancy, at an average of 267 days of gestation. Prepartum occurrence of LDA has previously been recorded in heifers and cows during their second and third pregnancies [2,4,16]. LDAs can develop in pregnant cattle between 16 weeks before calving and the end of pregnancy [2,3,5,6,17]. Given that most pregnant bovine LDA cases occur within three weeks of calving [2], the present cases were representative of bovine cases involving LDAs during pregnancy.

The rolling technique, which contributes to the conservative correction of LDA, has been suggested as a risk factor to induce torsion of the pregnant uterus if this technique is conducted within six to eight months of gestation [10]. This prediction was demonstrated based on a recent clinical trial in which this technique resulted in the involvements of uterine torsion and abortion in 20 and 10% of the pregnant cases, respectively, when treated within three to eight months of gestation [11]. Therefore, this technique is not recommended for correction of LDA during late pregnancy. In a limited number of clinical bovine reports, laparotomy was the predominant technique chosen for surgical correction of LDA during pregnancy [2,4,6,18]. It may be difficult to reposition the abomasum through right-flank laparotomy given that a large gravid uterus typically prevents the procedure during late pregnancy [18]. Therefore, left-flank or two-flank laparotomy is recommended, given that both can contribute to the diagnosis and treatment of LDAs during late pregnancy [2,4,6,18]. Right-paramedian abomasopexy may not be applicable to pregnant cows with LDAs, as these cases require dorsal recumbency if chemically restrained [13]. The blind suture technique, which allows minimally invasive surgical correction of LDAs, may not be appropriate for the treatment of LDAs in pregnant cows because of the risk of accidental injury to the gravid uterus [19]. To correct LDA using the TPS technique, the cow should be laid in dorsal recumbency while rolling clockwise [12,14]. However, this procedure can cause physical complications, including abortion, stillbirth, and iatrogenic torsion of the uterus, in pregnant cases [10,11].

Since the 2000s, laparoscopy has been used to assist the TPS technique in the treatments of LDAs [13]. Laparoscopic abomasopexy can be classified into two-step and one-step techniques. In the two-step technique, the cow is restrained in dorsal recumbency to perform the procedure; the suture of a TPS device is secured in the area of the right paramedian skin when the toggle bar is placed into the lumen of the abomasum [13]. The change from a standing to a recumbent position of the treated animals may have a potential to facilitate injury of the gravid uterus during the performance of the two-step laparoscopic abomasopexy. The one-step laparoscopic technique can be performed with the cows standing [13]. Additionally, the left flank is the proper surgical site for introducing trocars in this technique, allowing the replacement of the abomasum without being hindered by the gravid uterus [12]. Surgery durations using this study’s technique were between 15 and 40 min, which are similar to those required for one-step laparoscopic abomasopexy or two-step laparoscopic abomasopexy (15–20 and 27.5 min, respectively) [15,20]. The similar surgical durations were due to the gravid uterus no longer being an obstacle to the laparoscopic observation of the dislocated abomasum.

A variety of pregnancy rates in bovine cases treated with surgical corrections of LDAs have been reported, from lower values of between 35% and 42% [13,21] to higher values of 79.6% [22]. By 150 and 320 days postpartum, the pregnancy rates calculated from 13 of the present cases for which the reproductive records were available were 76.9% and 100.0%, respectively; these are higher than the corresponding pregnancy rates of 54.9% and 76.8% in animals in which LDAs were corrected using the TPS technique [23]. Additionally, the pregnancy rates at 90 days postpartum in the present cases (38.5%) were higher than those in animals treated with right paralumbar abomasopexy (30.0%) [22]. However, the pregnancy rates at 180 days postpartum did not differ between the present cases and the animals treated with right paralumbar abomasopexy (76.9% vs. 78.8%) [22]. The shortening of open days can partly depend on the length of the intervals between calving and the first service [22,23,24]. The intervals between calving and the first service in the LDA cases tended to be longer than those in animals kept in the same herds; the values were 87.1 days vs. 45.7 days in one report, respectively [24], and they were 90.9 days vs. 78.2 days in another report [23]. In the dairy farms where the present cases were kept, artificial insemination (AI) was commonly initiated during the first or second estrus exhibited after calving; the common interval between calving and the first service was <60 days after calving. This was followed by an ultrasonographic diagnosis of pregnancy one month after AI. After unsuccessful insemination, AI was then routinely performed in animals that exhibited estrus, as induced by prostaglandin administration. Repeating the AI procedure until conception may have increased the rate of service per conception and may have shortened the open days. Reproductive performance seems to be typically improved by 150 open days in animals treated with TPS and two-step laparoscopic abomasopexy [17,23,25]. Sick cows can suffer from a negative energetic balance due to the dramatic decrease in feeding and digestive disturbances in early lactation periods, during which LDAs are commonly present [23,25]. A negative energetic balance can delay normal ovarian activity [23,25]. Reproductive impairment resulting from a nutrient deficit could not have occurred in the present cases, in which LDAs were already corrected prior to calving, although a negative energetic balance could not be evaluated via blood biochemistry [15]. Uterine diseases, such as retained placenta, metritis, and endometritis, were not identified in the present cases, but have been reported to be predominantly found in 36.0%, 11.3–56.9%, and 39.6% of the postpartum diseases concurrent with or associated with LDA, respectively [16,17,26]. Additionally, these uterine diseases may be a confounding condition induced by surgery [26]. Pathological uterine conditions may result in delays within the estrous cycle, anestrus, and a decreased conception rate in LDA cases after calving. The corrections of LDAs in late pregnancy during this study had no negative associations with reproductive performance.

The one-year survival rates of animals treated with TPS have been reported to be between 40.5% and 62% [21,22]. A decrease in this rate is strongly dependent on increased culling rates [13]. In addition to the involvement of various postpartum diseases, such as mastitis, poor bovine productivity, including low milk yields and infertility, is a common determinant of culling [13,17]. In this study, the good reproductive performance may have contributed to the longer breeding period of the present cases. 

Several favorable conditions could have contributed to the positive outcomes in the present cases. Abomasal adhesion is frequently noted as one of the complications associated with LDA, accounting for approximately 10% of all complications [8]. Abomasal adhesions are associated with fibrous peritonitis in prepartum LDA [2,6]. Of the 15 animals treated with one-step laparoscopic abomasopexy, three experienced a recurrence of LDA that had been previously corrected with right-flank omentopexy. Previous surgical therapy could have caused abomasal adhesions, making dissection between the abomasum and adjacent structures difficult [2,6]. This would have hindered one-step laparoscopic abomasopexy if the present cases involved such adhesions, given that the dissection of abomasal adhesions is particularly difficult, even if conducted manually during laparotomy [2,6,8]. Although twin pregnancy did not occur in the present cases, it may obstruct the laparoscopic view [2]. Twin pregnancy can trap the dislocated abomasum between the rumen and the extremely gravid uterine horn [2]. With this in mind, prior to the application of one-step laparoscopic abomasopexy for pregnant animals with LDAs, surgeons should consider whether unsuitable conditions, such as twin pregnancy and abomasal adhesion, are present. Nevertheless, if unsuitable conditions are discovered during the performance of one-step laparoscopic abomasopexy, a rapid intraoperative switch to left- and/or right-flank laparotomy is recommended [2].

## 5. Conclusions

A standing position may help prevent postoperative complications, such as torsion and abortion, in pregnant cows when treated with one-step laparoscopic abomasopexy. The surgical route from the left flank in this technique is suitable for the correction of LDAs in pregnant cows, as the gravid uterus is not a total obstacle to the observation of LDAs. The positive postoperative results reported here demonstrate the efficacy of one-step laparoscopic abomasopexy in pregnant bovine cases with LDAs.

## Figures and Tables

**Figure 1 animals-12-03264-f001:**
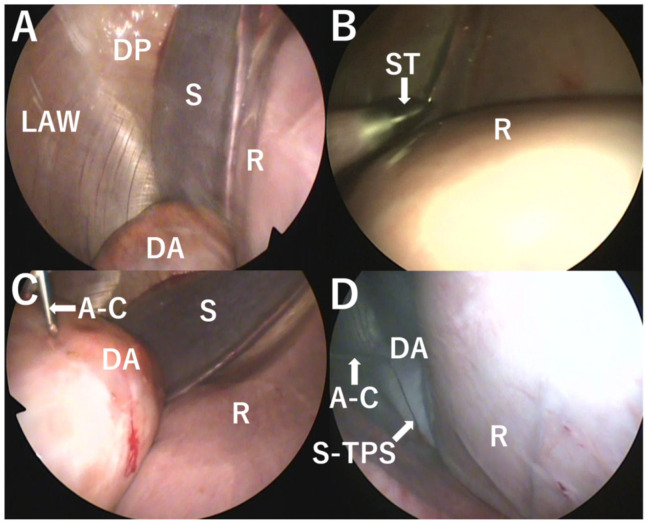
Laparoscopic views showing the cranial left abdominal cavity from an endoscope passing through a first trocar. (**A**) The displaced abomasum (DA) is seen in the space between the left abdominal wall (LAW) and the rumen (R). S: spleen; DP: diaphragm. (**B**) The apex of a second trocar (ST) is seen in the left abdominal cavity when being introduced into the 11th and 12th intercostal spaces. R: rumen. (**C**) The apex of a cannula (A–C) is advanced toward the DA while being inserted through the ST. S: spleen; R: rumen. (**D**) The suture part of a toggle–pin–suture (TPS) device (S-TPS) is evident between the DA, in which the bar part of the TPS device is secured into the lumen, and the apex of the cannula (A–C) when being pulled out from the lumen of the DA. R: rumen.

**Figure 2 animals-12-03264-f002:**
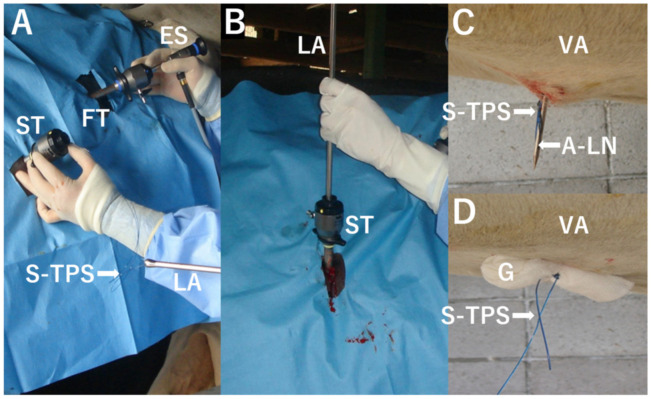
Intraoperative photos of laparoscopic abomasopexy. (**A**) An endoscope (ES) is inserted through a first trocar (FT) introduced into the left flank close to the 13th rib. The suture part of a toggle–pin–suture device (S-TPS), secured to an apex of the needle withdrawn into a long applicator (LA), comes out of a second trocar (ST), introduced into the 11th and 12th intercostal spaces. (**B**) The long applicator (LA) passing through the second trocar (ST) is operated so that the apex of the LA can be advanced toward the ventral abdominal wall under endoscopic observation. (**C**) The apex of the long needle (A-LN), in which the S-TPS is secured, is protruding outside of the ventral surface of the abdomen (VA). (**D**) The S-TPS is tied over a gauze stent (G) in the VA.

**Table 1 animals-12-03264-t001:** Summary of the clinical records of the present cases on the operative days.

Cow No.	Farm (Floor Type) ^a^	Age (day)	Parity	Gestation Day
1	A (T)	1642	1	273
2	A (T)	1889	2	274
3	B (F)	2209	3	268
4	B (F)	2408	3	268
5	C (T)	1931	2	260
6 ^b^	C (T)	2069	3	265
7	D (T)	1333	1	267
8	E (T)	1135	1	274
9	F (T)	2023	2	262
10	G (T)	2245	3	271
11 ^b^	H (T)	2604	5	269
12	I (T)	2400	4	263
13	J (F)	1770	2	259
14 ^b^	J (F)	1570	2	264
15	K (T)	783	0	266
Average (SD)		1867.4 (506.7)	2.3 (1.3)	266.9 (4.8)

^a^ Floor types of 11 farms (A to K) are abbreviated as T for tie-stall housing and F for free-stall or free-barn housing. ^b^ History of treatment with right-flank omentopexy during the prior postpartum periods was recorded.

**Table 2 animals-12-03264-t002:** Postoperative performance in the present cases.

Cow No.	Postoperative Day to Delivery	Postpartum Disease	Services per Conception	Day Open	Interval between Surgery and Culling	Cause of Culling ^e^
1	13		3	128	617	Unknown
2	10	Arthritis	0 ^a^	-	443	PA
3	17	Mastitis	4	154	671	Unknown
4	17		5	181	517	Unknown
5	15		NR ^b^	91	155	HD
6	13	Sole ulcer	1	78	118	Sole ulcer
7	19		1	63	83	PMP
8	11		2	59	962	Unknown
9	23		3	74	369	Unknown
10	14		3	132	204	Unknown
11	18		0 ^a^	-	420	PQM
12	22		4	122	157	Unknown
13	27		2	104	142	Unknown
14	23		4	160	487	Unknown
15	19		3	90	884	PQM
Average (SD)	17.4 (4.9)		2.9 (1.2) ^c^	110.5 (39.1) ^d^	415.3 (279.8)	

^a^ Artificial insemination was not performed because of the planned culling during the lactation period. ^b^ Not recorded. ^c^ The value was acquired based on the records from 12 animals, except for Nos. 2, 5, and 11. ^d^ The value was acquired based on the records from 13 animals, except for Nos. 2 and 11, in which culling has been planned. ^e^ PA, periarthritis; HD, hip dislocation; PMP, poor milk productivity; PQM, poor quality of milk.

## Data Availability

The data presented in this study are available on request from the corresponding author.

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
