# Peer review of "One-Step Laparoscopic Abomasopexy for Left Displacements of the Abomasum in Pregnant Cattle: A Retrospective Study"

_animals, 2022, doi:10.3390/ani12233264_

Round 1
Reviewer 1 Report
The manuscript provides interesting information on the follow-up of 1-step laparoscopic abomasopexy in dairy cattle. The English needs to be edited to improve the readability of the manuscript. The discussion also needs to be revised. It should discuss high points and major findings of the study by referencing findings without actually repeating the results.
Author Response
Thank you for your providing kindly advices. The corrected and added parts are highlighted by yellow boxes in the revised manuscript.
Question: The English needs to be edited to improve the readability of the manuscript.
Answer: The revised version is corrected by another English proof-reading company (Editage English editing service).
Question: The discussion also needs to be revised. It should discuss high points and major findings of the study by referencing findings without actually repeating the results.
Answer: According to the advice, the total parts of the revised version are improved by the addition of six reference papers.
Reviewer 2 Report
In order to perform an appropriate review of the content I would ask for this manuscript to be sent back for revision of English language. Some phrases the authors used don't sound appropriate, and some paragraphs lack a natural flow; in some areas the lack of thorough understanding of the English language is likely to cause misunderstanding or confusion for the reader; this is predominantly an issue of the correct use of medial and farming/ cow related term. Some examples from the summary/ abstract and introduction include: Simple Summary: Line 13: "several tens of minutes" is not an informative term, please be more concise or remove this description entirely Line 14/15: "good reproductive performance over their lactation period" that sounds like you are referring to milk production not reproductive performance? Line 15: "reared" is not the correct English term here unless you are talking about heifers; rearing is used for growing young animals; Abstract: Line 19: replace "invest" with "investigate" ; same line: it is not clear what you mean by "status" in this context Line 27: I would not consider arthritis or sole ulcer a typical post-partum disease; please rephrase this sentence Introduction: Line 38: be more concise about the period Lie 43-45: Please revise the ethology of the development of LDAs using more recent literature Line 49; "Cease of faeces" please revise English wording
Author Response
Thank you for your providing kindly advices. The corrected and added parts are highlighted by yellow boxes in the revised manuscript.
Question: In order to perform an appropriate review of the content I would ask for this manuscript to be sent back for revision of English language. Some phrases the authors used don't sound appropriate, and some paragraphs lack a natural flow; in some areas the lack of thorough understanding of the English language is likely to cause misunderstanding or confusion for the reader; this is predominantly an issue of the correct use of medial and farming/ cow related term.
Answer: The revised version is corrected by another English proof-reading company (Editage English editing service). Additionally, the total parts of the revised version are improved by the addition of six reference papers.
Question: Summary: Line 13: "several tens of minutes" is not an informative term, please be more concise or remove this description entirely
Answer: The terms "several tens of minutes" are deleted.
Question: Line 14/15: "good reproductive performance over their lactation period" that sounds like you are referring to milk production not reproductive performance?
Answer: In the sentence, the terms “over their lactation period" are deleted.
Question: Line 15: "reared" is not the correct English term here unless you are talking about heifers; rearing is used for growing young animals.
Answer: The term “rear” is replaced by “breed (bred)” or “maintained”, throughout the revised manuscript.
Question: Line 19: replace "invest" with "investigate" ; same line: it is not clear what you mean by "status" in this context.
Answer: The term "invest" is corrected as "investigate". Additionally, the terms “the states and” are deleted.
Question: Line 27: I would not consider arthritis or sole ulcer a typical post-partum disease; please rephrase this sentence.
Answer: It is well-known that postpartum involvements of sole ulcer is related to thinning in the digital cushion associated with consumption of fat tissues after calving. According to Reviewer2’s advises, in the revised version, this sentence is corrected.
Question: Line 38: be more concise about the period.
Answer: According to the description in Pardon’s report, this sentence is corrected.
Question: Lines 43-45: Please revise the ethology of the development of LDAs using more recent literature.
Answer: The etiology to develop LDA in pregnant cows seems to be fully un-known based on our reading the past and recent correlating papers. The sentences about the etiology are minorly corrected by addition of one paper (Pardon et al., 2012).
Question: Line 49; "Cease of faeces" please revise English wording
Answer: The term “Cease of feces” is replaced by "constipation".
Reviewer 3 Report
The article has a high professional level, it brings new therapeutic procedures. The results bring new knowledge that can be used in veterinary practice and in science.
I recommend publishing the work.
Author Response
Thank you for praising our manuscript. According to Reviewer1’ and Reviewer2’ comments, the revised version is corrected in the parts highlighted by yellow boxes. Additionally, the revised version is corrected by another English proof-reading company (Editage English editing service).
Round 2
Reviewer 1 Report
The reviewer would like to thank the authors for the manuscript and the edits they provided in the revision. There are a a few more changes that would make the publication stronger listed below:
Lines 11-12: Delete "because of the good therapeutic efficacy of this technique.
Line 12: This technique allows for appropriate correction...
Line 14: replace "before" with prior to"
Line 14: Replace "could be" with "were successfully"
Line 16: Although technical description is published,...
Line 21: You say that the animals were non-sedated but state in the M&M that they were given 0.0.5mg/kg xylazine IV
Line 25: No cases experienced postoperative complications, and all had a normal delivery...
Line 35: Delete "The" before "displacement of the abomasum"
Line 38: Delete "between"
Line 75: Do you mean that the one-step laparoscopic technique is the most suitable laparoscopic technique? An argument could be made for left flank laparotomy. For example, as you state later, if adhesions are present.
Line 103 and 108: Changed "extended abomasum" to "distended abomasum"
Line 103-104: Change to "commonly detected in the space between the surface..."
Line 113: Replace "shrivel" with "decompress"
Line 135: How did you confirm? Via visualization with the endoscope? Or was it by identifying a bulge externally?
Line 138: I assume the gauze stent was applied externally by an assistant?
Line 157: Replace "disappeared" with "resolved"
Line 169: "Two additional animals..."
Line 181: Change to "Fifteen 6-year-old, multiparous..."
Line 194: Therefore, this technique is not recommended...
Line 195-196: Delete "Despite no previous evidence... the body's rolling."
Line 196-209: This seems redundant. It is information from the introduction. If more detail is desired it should be placed in the intro.
Line 216: You used sedation
Line 219-220: Delete "the surgeons noted little... when applying this technique"
Line 264-267: This sentence needs revised. I am not sure what it is saying.
Line 292: "such as torsion and abortion"
Author Response
Thank you for your continuous providing the kindly comments. The corrected parts are highlighted by yellow boxes.
Question: Lines 11-12: Delete "because of the good therapeutic efficacy of this technique.
Line 12: This technique allows for appropriate correction...
Line 14: replace "before" with prior to"
Line 14: Replace "could be" with "were successfully"
Line 16: Although technical description is published,...
Answer: According to above-mentioned advices, these sentence are corrected.
Question: Line 21: You say that the animals were non-sedated but state in the M&M that they were given 0.0.5mg/kg xylazine IV
Answer: According to the comment, “on non-sedated animals” are deleted in this sentence.
Question: Line 25: No cases experienced postoperative complications, and all had a normal delivery...
Line 35: Delete "The" before "displacement of the abomasum"
Line 38: Delete "between"
Answer: According to above-mentioned advices, these sentence are corrected.
Question: Line 75: Do you mean that the one-step laparoscopic technique is the most suitable laparoscopic technique? An argument could be made for left flank laparotomy. For example, as you state later, if adhesions are present.
Answer: This sentence is deleted in revised manuscript.
Question: Line 103 and 108: Changed "extended abomasum" to "distended abomasum"
Line 103-104: Change to "commonly detected in the space between the surface..."
Line 113: Replace "shrivel" with "decompress"
Answer: According to above-mentioned advices, these sentence are corrected.
Question: Line 135: How did you confirm? Via visualization with the endoscope? Or was it by identifying a bulge externally?
Answer: Assistant person could macroscopically observe move of the skin surface by pushing with the long needle from the abdominal cavity. However, under endoscope view, surgeon could not see whether the apex of the long applicator could reach for the recommended position. Thus, in the revised version, “it was confirmed that” are deleted in this sentence.
Question; Line 138: I assume the gauze stent was applied externally by an assistant?
Answer: Yes. Thus, “by assistant veterinarian” are added in this sentence.
Question: Line 157: Replace "disappeared" with "resolved"
Line 169: "Two additional animals..."
Line 181: Change to "Fifteen 6-year-old, multiparous..."
Line 194: Therefore, this technique is not recommended...
Line 195-196: Delete "Despite no previous evidence... the body's rolling."
Answer: According to above-mentioned advices, these sentence are corrected.
Question: Line 196-209: This seems redundant. It is information from the introduction. If more detail is desired it should be placed in the intro.
Answer: In Discussion section, when these lines are deleted, structure of text is not good. Thus, the lines 68-72 “Regarding the rolling technique applied in pregnant cases, … as the gravid uterus prevents observation of the displaced abomasum from the surgical opening” are deleted in Introduction section of the revised version. Additionally, the lines 74-77 “The one-step laparoscopic technique is presumed … from the left flank is the recommended surgical route [2]” are also deleted.
Question: Line 216: You used sedation
Answer: In the revised version, “without sedation or” are deleted in this sentence.
Question: Line 219-220: Delete "the surgeons noted little... when applying this technique"
Answer: According to the comment, this sentence is corrected.
Question: Line 264-267: This sentence needs revised. I am not sure what it is saying.
Answer: According to the comment, the sentences about postsurgical recurrence (lines 267-271) are deleted in revised version.
Question: Line 292: "such as torsion and abortion"
Answer: According to the comment, this sentence is corrected.
Reviewer 2 Report
This manuscript has improved after extensive revisions. Some areas still need a small amount of work but then it should be suitable for publication
Line 28 : “Fifteen animals were maintained after 415.3 ± 279.8 postoperative days” – this is a bit misleading, I would rather insert one of your other statements “ The one year survival rate was 60% (9/15)”
Line 47-49: Please rephrase this sentence to make it more clear and readable
Line 94-109: How was visibility of organs achieved for these endoscopic procedures (insufflation)? Air, N(2)O or CO2?
Line 101: why bluntly?
General comment for point 2.2 Procedure of one-step laparoscopic abomasopexy for pregnant cattle:
Please be concise when you use “a second trochar” or if you mean “the second trochar” which is the same trochar you have already described. For example, should line 126 read “the” instead of “a” or did you use a new trochar?
Table 1: 3 out of 15 cows had already had surgical intervention for LDA, that seems very high percentage of relapse! Please discuss
Line 178: Please correct, not all 15 cows were 6 years old!
Line 208-211: please be more precise, the toggle is first inserted endoscopically from the left flank on the standing cow and THEN she is restrained in dorsal recumbency to secure the suture to the ventral abdominal wall; sedation is not required for this technique is you have a suitable tilt table available; but I agree with the statement that this technique is not suitable for end-term pregnant cattle
Line 242-245: please rephrase this paragraph to make it clearer: so say there is sufficient dietary intake but a dramatic decrease in feeding, that doesn’t make sense
Author Response
This manuscript has improved after extensive revisions. Some areas still need a small amount of work but then it should be suitable for publication
Thank you for your repeatedly providing the helpful advices. In the revised version, the corrected parts are highlighted by green boxes.
Question: Line 28 : “Fifteen animals were maintained after 415.3 ± 279.8 postoperative days” – this is a bit misleading, I would rather insert one of your other statements “ The one year survival rate was 60% (9/15)”
Answer: According to this comment, the sentence “Fifteen animals were maintained after 415.3 ± 279.8 postoperative days” is replaced by the sentence “The one year survival rate was 60% (9/15)”.
Question: Line 47-49: Please rephrase this sentence to make it more clear and readable.
Answer: In the revised version, this sentence is corrected.
Question: Line 94-109: How was visibility of organs achieved for these endoscopic procedures (insufflation)? Air, N(2)O or CO2?
Answer: During our procedure of laparoscopic abomasopexy, intra-abdominal insufflation is performed only when poor visibility on endoscopic observation. Thus, the use of intra-abdominal insufflation is decided during surgery. When laparoscopic abomasopexy has applied for fifteen pregnant cows used in this study, use of intra-abdominal insufflation has not been required based on the good laparoscopic observations in all animals.
Question: Line 101: why bluntly?
Answer: According to this comment, a term “bluntly” is removed in this sentence. Additionally, in the legend of Figure 1, a term “bluntly” is also removed.
Question: Please be concise when you use “a second trochar” or if you mean “the second trochar” which is the same trochar you have already described. For example, should line 126 read “the” instead of “a” or did you use a new trochar?
Answer: According to this suggestion, which of “a” and “the” is chosen in paragraph about laparoscopic abomasopexy’s procedure and in Figures 1 and 2
Question: Table 1: 3 out of 15 cows had already had surgical intervention for LDA, that seems very high percentage of relapse! Please discuss.
Answer: In our field, 1 % of Holstein cows involve LDAs during their pregnant periods. As most cases have no or slight clinical signs, the common therapeutic choice for these cases is either waiting its pregnancy without treatment, medication for gastrointestinal symptoms, or pregnancy induction. We always decide use of laparoscopy for correction of the displaced abomasum, when the pregnant cases have severe clinical signs. Additionally, we perform laparoscopic abomasopexy initiatively for the pregnant cases if having history of recurrence. Thus, our therapeutic strategy may contribute to elevated percentage of laparoscopic abomasopexy applied for the recurrent, pregnant cases.
Question: Line 178: Please correct, not all 15 cows were 6 years old!
Answer: In the revised version, “6 year old” is deleted in this sentence.
Question: Line 208-211: please be more precise, the toggle is first inserted endoscopically from the left flank on the standing cow and THEN she is restrained in dorsal recumbency to secure the suture to the ventral abdominal wall; sedation is not required for this technique is you have a suitable tilt table available; but I agree with the statement that this technique is not suitable for end-term pregnant cattle
Answer: We agree the Reviewer 2’s comment. Because we have used small dose of xylazine (slight sedative level) in our procedure of laparoscopic abomasopexy, the sentence “Thus, the two-step laparoscopic technique, in which sedation is required, seems to be unsuitable for pregnant cows with LDAs” is deleted. Through this sentence, we hope to describe that the change in standing to recumbent position of the treated animals may have a potential to facilitate injury of the gravid uterus during performing two-step laparoscopic abomasopexy. Thus, the new sentence is added here in the revised version.
Question: Line 242-245: please rephrase this paragraph to make it clearer: so say there is sufficient dietary intake but a dramatic decrease in feeding, that doesn’t make sense.
Answer: In the revised version, “Despite sufficient dietary intake to increase milk production” is deleted, because these terms were the cause of confusing.